# Analysis and Synthesis in the Design of Magnetic Switching Electric Machines

**Nikolay Shaitor [1], Michal Kelemen [2],* and Boris Yakimovich [1]**

[1] Department of Renewable Energy Sources and Electrical Systems and Networks,
Institute of Nuclear Energy and Industry, Sevastopol State University, Kurchatova 7,
299015 Sevastopol, Republic of Crimea, Ukraine; shaytor1950@mail.ru (N.S.); yakimovich52@gmail.com (B.Y.)

[2] Department of Mechatronics, Faculty of Mechanical Engineering, Technical University of Kosice, Letna 9,
040 01 Kosice, Slovakia

\* Correspondence: michal.kelemen@tuke.sk; Tel.: +421-556022388

**Abstract:** A systematic approach to the design of electrical machines is implemented by solving problems of analysis and synthesis in various combinations at different stages and stages of design. The questions of the formulation and implementation of synthesis and analysis problems in the study and design of modular-type magnetically commutated electrical machines are considered. They are aimed at reducing weight, size and cost while improving the performance of these newly designed machines. A complex method of parametric synthesis and an automated program containing a calculated mathematical model of an electric machine has been developed. On the basis of numerical full factorial experiments, the optimization parameter is determined, and a regression analysis is performed with the construction of an optimization model. It allows you to find a narrow range of variation of significant factors, at which the optimization parameter satisfies the specified conditions. On the example of studying an inductor generator of axial-radial configuration, new approaches to the formulation and solution of typical problems of analysis and synthesis of modular-type electrical machines are shown. The use of complex parametric synthesis makes it possible to significantly reduce the masses of the designed modular machines in comparison with drum-type inductor machines of the same power.

**Keywords:** design optimization; specific power; electromagnetic module; calculation model; factor analysis; gradient method





## 1. Introduction

The practice of carrying out design work is the alternation of solving problems of analysis and synthesis. The mathematical support of the process of solving these problems is made up of various models of electrical machines and algorithms for solving direct, inverse and optimization problems [1]. The optimized design of electrical machines is a nonlinear multi-objective problem. A substantial amount of literature has been devoted to this subject over the last decade [2]. In electrical machine design, objectives such as highest efficiency, lowest cost and minimum weight of active materials have to be simultaneously met [3]. Design, simulation and diagnostics of electrical machines is a key task mainly in the fields of automation, mechatronics, robotics and the automotive industry [4–13].

The idea of creating a magnetically commutated electric machine belongs to Professor A. A. Afonin [14]. In his works, questions of the design are stated, an exhaustive analysis of electromagnetic processes is carried out, the numerous advantages of such machines are shown, and the areas of their application are determined. The results of these studies can serve as a basis for the design and calculation of the machine [15]. Considerable attention is paid to analytical and model research of modular electrical machines [16]. Specialists and researchers are attracted by the reliability, manufacturability and ability to transform configurations when designing these machines for various applications [17]. By their

design, they are referred to as modular machines, and by the principle of operation, as synchronous reluctance-inductor machines [18].

However, the creation of a series of electrical machines or a separate machine, optimal in all respects, is practically impossible [19]. The optimal design of electrical machines is reduced to a general nonlinear programming problem [20]. The mathematical model of the main problem of nonlinear programming, covering the design of electrical machines, can be represented as

$$
\begin{aligned}
&F(X_1, X_2, \ldots, X_p) = \min; \\
&G_1(X_1, X_2, \ldots, X_p) \geq 0; \\
&G_2(X_1, X_2, \ldots, X_p) > 0; \\
&-\cdot-\cdot-\cdot-\cdot-\cdot-\cdot-\cdot-\cdot-\cdot-\cdot \\
&G_m(X_1, X_2, \ldots, X_p) \geq 0; \\
&X_i \geq 0; \quad i = 1, 2, \ldots, p.
\end{aligned}
\tag{1}
$$

The design of an electric machine is reduced to multiple calculations of dependencies between the main indicators, given in the form of a system of formulas, empirical coefficients, graphical dependencies, which can be considered as design equations [21]. There is currently no universal method for solving such a problem. However, there are a large number of well-developed methods for solving optimal problems of a certain structure. In the design of magnetically commutated electric machines, the methods of factor analysis and gradient methods have proven themselves well.

The method of factor analysis [22] provides the approximation of the objective function and expressions for limiters by polynomial dependences in the form

$$
f(X_1, \ldots, X_k) = b_0 + \sum_{i=1}^{k} b_i X_i + \sum_{j<i}^{k} b_{ji} X_i X_j + \sum_{i=1}^{k} b_{ii} X_i X_i^2 + \ldots .
\tag{2}
$$

Such dependences are called response functions or response surfaces, and the variables $X_1, \ldots, X_k$ are called factors, the analysis of which is carried out in the $k$-dimensional factor space. The initial information for constructing the response functions is formed as a result of various combinations of variable values (factors), each of which occupies one of three levels: lower, upper or middle (base). The coefficients included in the response function are called regression coefficients. To simplify the analysis, some of the terms are excluded from the regression equation; however, the remaining terms must provide the necessary accuracy of the approximation. The polynomials obtained in this way are used in optimization algorithms.

Gradient methods are reduced to the process of successive approximations to the optimum. Their use is advisable when the function to be minimized (or maximized) is specified not analytically but in a tabular or graphical manner. Additionally, when the equations obtained by equating the partial derivatives to zero are undecidable in radicals. The process of finding the optimum is inevitably stepwise. Gradient methods differ from conventional stepwise search methods in that the step size depends on the rate of change of the function, which in some cases significantly speeds up the search process [23].

The aim of the work is the application of problems of analysis and synthesis in the design of modular-type magnetically commutated electric machines, which is aimed at reducing weight, dimensions and cost while improving the performance of machines of this design.

Electrical machine design methods are mainly oriented towards traditional synchronous and asynchronous machines. These machines contain distributed electrical stator windings in a lumped magnetic system. A distinctive feature of the considered electrical machines is concentrated windings with a distributed magnetic system consisting of separate modules. As a result, the development of methods for the design of modular magnetic switching machines refers to the solution of urgent problems [24].

## 2. Materials and Methods

To achieve the goal, an integrated method of parametric synthesis was developed based on a combination of an optimization procedure based on the results of a full-factor numerical experiment, with the synthesis in the direction from electromagnetic loads to the dimensions of the machine. The input quantities are electromagnetic loads, and the output quantities are the geometrical dimensions of the machine and the level of the optimization parameter. The output parameters are obtained by computer calculations using the method of factor analysis. The narrowing of the search area is carried out using the gradient method [25]. It is implemented in the form of an optimization computer program, where the input data are the results of a number of numerical experiments obtained using the program for calculating the electromagnetic core of a modular-type electromechanical converter. In general, the modular machine code method refers to parametric methods. In this case, a complex optimization procedure is applied within the framework of one selected variant of the machine design, and only the electromagnetic loads are varied [26]. The issues of heat and ventilation calculation of the machine are outside the scope of this study. The complex procedure of parametric synthesis consists of four stages, at each of which the range of variation of significant factors is narrowed, as shown in Figure 1. The complexity of the algorithms within the individual blocks in Figure 1 made it difficult to apply the pseudocodes in the Materials and Methods section. Therefore, the authors considered it acceptable to provide an extension and mathematical support of this section by simply presenting the material in the following sections.

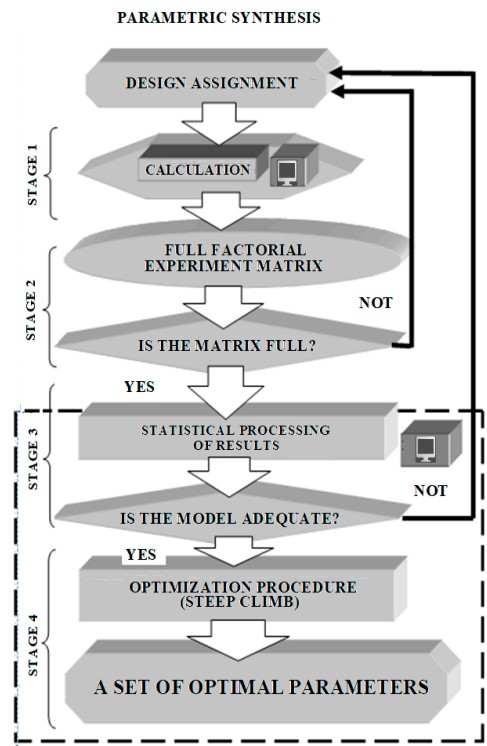

**Figure 1.** The procedure for complex parametric synthesis of a modular machine.

In the first stage, an automated program for calculating a modular machine is used. It implements synthesis from an incomplete dataset in the direction from electromagnetic loads to the size of the machine. At this stage, options that are not suitable for life are excluded [27]. The mathematical apparatus gives many numerical results, some of which have no physical meaning. For example, when solving cubic Equation (1), only the first real root has a physical meaning in Cardano's solution. Other roots have no physical meaning. In addition, some of the numerical results may go far beyond the design assignment. In

the course of calculations and analysis, the authors call such mathematical results unviable and exclude them from further consideration.

In the second stage, numerical experiments are carried out using an automated program for calculating a modular machine with the determination of optimization parameters. The calculation is carried out according to the matrix of a full-factor experiment, compiled on the basis of the theory of planning experiments [28]. As a result, the region of varying electromagnetic loads is found, which is close to the optimal one.

After filling in the matrix of the full-factor experiment, the transition to the third stage, which is built using a separate computer program. It implements statistical processing of the results of numerical experiments, determination of significant factors, regression analysis and an optimization model.

If an adequate model is obtained, the fourth stage is carried out, also implemented with the help of a separate computer program. This optimization procedure is based on the Box–Wilson gradient method. This allows you to find an even narrower range of changes in significant factors, at which the optimization parameter meets the specified conditions [29].

When carrying out an extreme computational experiment, the optimization parameter is a reaction (response) to the influence of factors that determine the behavior of the selected system. Certain requirements are imposed on the optimization parameter. It should be quantitative, measurable for the selected combination of factor levels, expressed as a single number and statistically unambiguous. The choice of the optimization parameter was made from the following considerations. As you know, traditional inductor machines, according to the principle of operation, have an incomplete use of magnetic flux. As a result, they have increased weight and dimensions compared to synchronous and asynchronous machines of the same power. Modular electromechanical converters are new induction machines designed for use in autonomous transport systems, as well as in conventional and renewable generation systems, where one of the main requirements is the minimum weight.

Therefore, the optimization parameter is a value associated with the minimum mass and minimum cost of the electromagnetic core of the machine. This value is the specific power by the mass of the electromagnetic core $p_{am}$, kW/kg. In other words, it is the active power per unit mass of only that structural part of the machine that contains magnetic and electrical circuits.

Numerical experiments of the first stage showed that the maximum specific power practically coincides with the minimum mass, volume and specific cost of the electromagnetic core, as well as with the maximum efficiency. In addition, the selected optimization parameter meets all the general requirements for optimization parameters. It is quantified and easily calculated during the synthesis procedure. In this case, the optimum of the objective function is associated with its maximum and corresponds to the problem of obtaining machines of a new type, which are distinguished by the increased specific power of the electromagnetic core in comparison with traditional induction machines.

## 3. Results

### 3.1. Features of the Design Model of an Electric Machine

The computational model of a magnetically commutated machine of modular configuration is used at the first stage of the complex parametric synthesis procedure shown in Figure 1. It is universal in relation to the design module and is applicable for various design options for the electromagnetic core, as shown in Figure 2.

A structural element of a magnetically commutated machine (elementary module) is shown in Figure 2. On the stationary part (stator), there are two Π-shaped inductors 1 (in Figure 2a, the inductors are turned with their ends outward). On the moving part (rotor), there is a part of the inductor 2, which closes the magnetic flux Φ. On the stationary part, there are two excitation windings 3, separated for each core. Direct currents in the field windings (I1, I2) flow in opposite directions. The armature winding 4 is located on the stationary part and passes through both inductors.

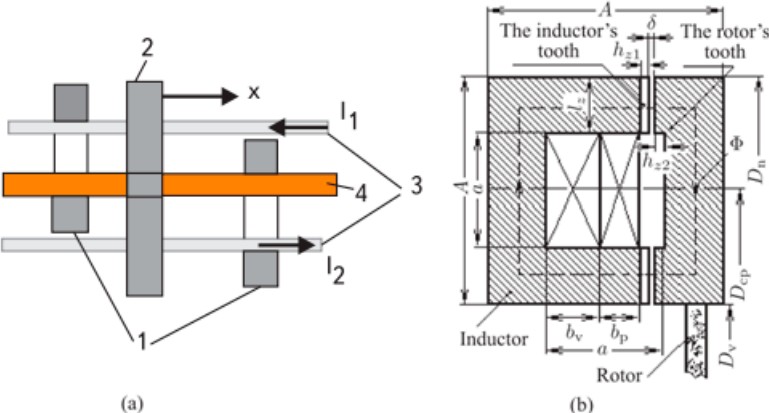

**Figure 2.** The elementary module of a machine with magnetic commutation: (**a**) principle of operation; (**b**) designations of geometric dimensions.

When moving (rotating) part 2 in the "X" direction, the flow is initially closed in the left inductor and then in the right one. Since the excitation currents flow in opposite directions, the magnetic flux covering the armature winding 4 changes sign. An alternating electromotive force is induced in the armature winding, and when an electrical load is connected to it, an alternating current flows. The machine is operating in generator mode.

Vice versa, if the armature winding is powered with alternating current, then an electromagnetic moment will act on the moving part of the machine. The machine operates in a motorized mode. If the frequency of the current in the armature winding is equal to $f = pn/60$ ($p$—the number of pole pairs, $n$—the rotation frequency), then the moment on the moving part will have some constant average value. The geometrical dimensions of the elementary module are indicated in Figure 2b, and the complete section of the machine is shown in Figure 3.

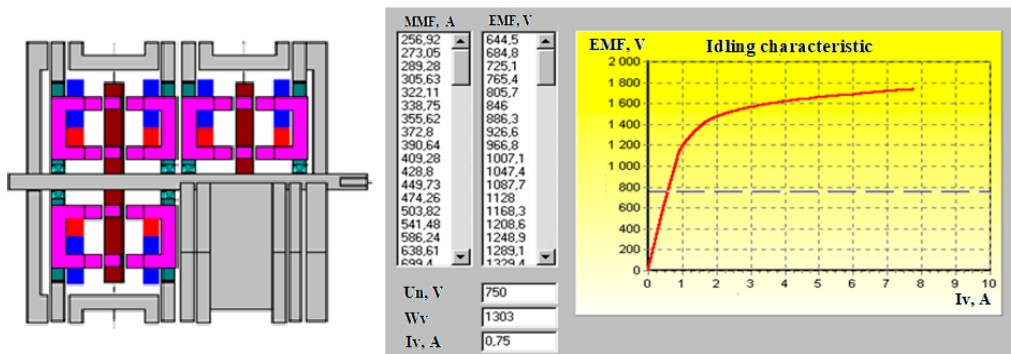

**Figure 3.** Sketch and characteristics of the generator.

The basic design equation of a modular machine is obtained based on the condition of the maximum magnetic flux of the module [30]. This equation has the form:

$$a^3 - aS_v - c = 0 \tag{3}$$

Here, $a$ is the size of the machine module for copper electrical windings; $c$—the effective volume of the working winding; $S_v$—the cross-sectional area of copper of the excitation winding:

$$S_v = \kappa_{pc}\kappa_{zp}k_F B_\delta \delta' / \mu_0 \kappa_{zm} j., \tag{4}$$

where $\kappa_{pc} = 1.5 \div 2.0$ is the scattering coefficient of the magnetic flux; $\kappa_{zp} = 1.15$ is the safety factor for the regulation of the generator's voltage; $k_F = 1.25 \div 1.45$ is the ratio of the magneto motive force of the magnetic circuit to that of the gap; $B_\delta = B_{z1} \kappa_{zc} \alpha_z$ is

the gap induction; $B_{z1}$ is the admissible value of induction in the teeth of the inductor; $\kappa_{zc} = 0.81 \div 0.99$ is the coefficient of filling with steel; $\delta' = 4(I - 1)\,\delta$ is the calculated value of the air gap ($i$ is the number of the axial layers or disks of the stator); $\mu_0 = 4\pi \cdot 10^{-7}$ H/m is the magnetic constant; $k_{zm} = 0.35 \div 0.65$ is the coefficient of filling the window with copper; $j$—the current density in windings.

The effective volume of the working winding is found from the expression:

$$c = [P\,\kappa_\lambda / 42, 62(\kappa_\lambda - 1)\,\kappa_{zp}\kappa_{zm}jb_z B_\delta] \times (n/f^2), \tag{5}$$

where $P = EI_n \cos\varphi_n$ is the calculated power given during the design; $E = (1.15 \dots 1.3)U_n$ is the EMF for the generator, and $E = (0.7 \dots 0.85)U_n$ is the EMF for the engine; $U_n$ is the rated voltage; $I_n$ is the rated current; $\cos\varphi_n$ is the rated power factor; $k_\lambda = \lambda_{max}/\lambda_{min} = 1.5 \div 2$ is the magnetic flux modulation index; $b_z = b_{z\,min} = (3 \div 5) \cdot 10^{-3}$ is the minimum permissible tooth width; $f$ is the frequency of electric current; $n$ is the frequency of the electric current; rotor speed.

When solving the main design Equation (3), it is necessary to take into account the following considerations. In the case, where $(c/2)^2 - (S_v/3)^3 \geq 0$, one should use the Cardano solution [31], in which only the first real root has a physical meaning:

$$a = \sqrt[3]{c/2 + \sqrt{(c/2)^2 - (S_v/3)^3}} + \sqrt[3]{c/2 - \sqrt{(c/2)^2 - (S_v/3)^3}} \tag{6}$$

Otherwise, you will have to resort to the trigonometric solution:

$$a = 2\sqrt{(S_v/3)}\cos(\alpha/3), \tag{7}$$

where $\alpha = \arccos\left[c/2\sqrt{(S_v/3)^3}\right]$ is a constant determined by the parameters of the electrical windings of the machine.

The cross-sectional area for copper of the working winding:

$$S_p = a^2 - S_v. \tag{8}$$

The dimensions of the other two sides of the field winding and the working winding, respectively:

$$b_v = S_v/a; b_p = S_p/a. \tag{9}$$

The leading dimensions (Figure 4) are determined from the outer diameter:

$$D_n = D_v + A. \tag{10}$$

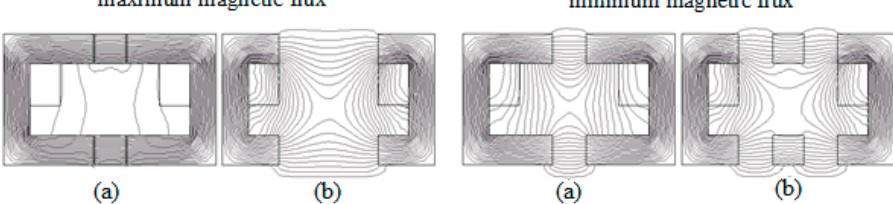

**Figure 4.** The magnetic field of the machine: (**a**) along the teeth; (**b**) along the grooves.

The axial dimension:

$$l = (i - 1)\lfloor 2(h_{z1} + \delta) + h_{z2}\rfloor + ia + 2l_z., \tag{11}$$

where $i$ is the number of disks of the stator; $h_{z1}$ and $h_{z2}$ are the axial dimensions of the teeth of the inductor and the rotor, respectively.

The total size of the core of the module $A$ and its width $l_z$ are set from the condition of the optimal ratio of copper and steel.

$$A = 1.61a; \qquad l_z = 0.32a. \tag{12}$$

Since the machine is designed so that the rated voltage, current, electric frequency and rotational velocity are taken to be given, and the gap width, gap induction and current density can be chosen in accordance with design recommendations, and the resulting Equations (3)–(12) uniquely define the leading dimensions of the machine [32].

The total volume of a modular machine with magnetic commutation is determined taking into account the outer diameter from Equation (10) and the axial dimension from Equation (11) of the machine:

$$V = 0.25\pi D_n^2 l. \tag{13}$$

The volume of the electromagnetic core (Figure 4) is made up of copper windings and steel, consisting of inductors and teeth:

$$V_{ea} = V_m + V_{ci} + V_{cz} + V_{cp}, \tag{14}$$

where $V_m = \pi D_{cp} a^2$ is the volume of copper windings; $V_{ci} = N \cdot 3 \cdot z \cdot b_i \cdot l_z \cdot a$ is the volume of the inductor steel; $V_{cz} = N \cdot 2 \cdot z \cdot h_{z1} \cdot b_{z1} \cdot l_z$ is the volume of the steel of the inductor teeth; $V_{cp} = 2z h_{z2} b_{z2} l_z$ is the volume of the rotor teeth steel.

Here, the average values of the width of the inductor $b_i$, the width of the teeth of the inductor $b_{z1}$ and the teeth of the rotor are taken $b_{z2}$, calculated from the average diameter of the inductor $D_{cp} = 0.5\,(D_v + D_n)$.

The mass of the electromagnetic core includes the mass of the copper of the electrical windings and the steel of the machine:

$$G_a = V_m g_m + (V_{ci} + V_{cz} + V_{cp})g_c, \tag{15}$$

where $g_m$ and $g_C$ are the specific indicators of the mass of copper and steel, respectively.

As a parameter for optimizing an electric machine, it is advisable to consider the maximum specific power, that is, the amount of power per unit mass of the machine:

$$p_{am} = P/G_a. \tag{16}$$

The program for calculating the inductor-modular machine is implemented in an integrated environment in the Delphi 10.3 Rio language [33]. The program structure consists of program blocks, as shown in Figure 5. The algorithm for calculating each block is based on the calculation Equations (2)–(16). The program allows you to enter and edit three groups of parameters.

The first group of parameters is the nominal parameters of the design task for calculating the machine: power; voltage of the armature winding and excitation winding; electrical frequency; rotation frequency; power factor; number of phases.

The second group of parameters are the values of independent design parameters that the developer chooses from the experience of designing electrical machines.

The third group of parameters includes the preliminary values of the calculated coefficients, which the developer accepts before calculating, based on the experimental data, and which the program refines in a recursive mode. The main calculated coefficients are the magnetic flux leakage coefficient $k_\delta$ and the modulation coefficient of the main flux $k_\lambda$.

First, the block of preliminary calculations of the module geometry works, the dimensions of which are indicated in Figure 4. After that, the stator tooth saturation check unit comes into operation. Here, the value of the magnetic induction in the most loaded part of the module is checked.

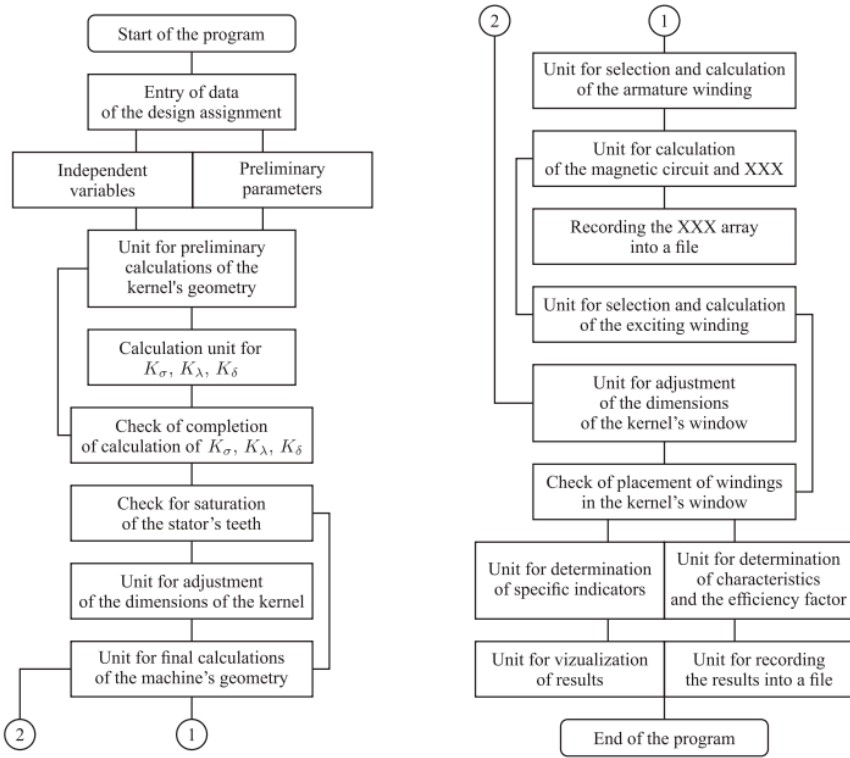

**Figure 5.** A flowchart of the calculation process.

If necessary, the section of the module tooth is corrected, the dimensions of the electromagnetic core are changed, and again, in the recursive mode, the coefficients $k_\delta$ and $k_\lambda$ are refined. In the next block, the geometric dimensions of the machine module are finally determined.

Next, the block for selecting and calculating the working armature winding comes into operation. It determines and specifies the number of turns, the cross-section of the conductors, taking into account the insulation, the area of the module window for the working winding.

After selecting the armature winding, the program proceeds to the block for calculating the magnetic circuit and the idling characteristics. The magnetization curves $B = \mu$ (H) for the selected steel grade are given in an analytical form, and the calculation is carried out as a result of referring to the appropriate procedure.

Based on the obtained geometry data of the computational domain, a numerical solution of the field problem by the finite element method is carried out using the standard FEMLAB 3.0 package from COMSOL [34]. The results of calculating the magnetic field in the main sections, along the teeth and grooves of the machine, are shown in Figure 4.

The results from calculating the magnetic field make it possible to refine the preliminary results of the analytical calculation of the scattering coefficients of the magnetic flux $k_\delta$ and its modulation $k_\lambda$.

The program of this block is made in accordance with the set value of the induction in the air gap. Based on the results of comparing the current value of induction with a given value, a transition to the block for selecting and calculating the field winding occurs.

In this block, the cross-section of the excitation winding copper and the number of turns in the excitation coil of one module are determined. Then the field current and wire diameter are determined. In conclusion, the area of the window occupied by the insulated wires of the working winding and the field winding is calculated.

After the windings are selected, the program checks the placement of the insulated wires of the windings in the area of the module window and, if necessary, adjusts its

dimensions. If such a need arose, the program returns control to the block for final calculations of the geometric parameters of the machine.

In the block for determining specific indicators, the mass of active materials of the electromagnetic core is calculated. This block calculates optimization parameters such as specific gravity, power and cost of a modular inductor machine.

The unit for determining the efficiency makes it possible to determine this indicator according to well-known methods for traditional electrical machines. In this case, losses in copper and steel are calculated taking into account the temperature, as well as a function of the current frequency, taking into account the surface effect.

In the block visualization of results, the program drawing of a sketch of a modular machine in a given scale is carried out, as well as its characteristics are depicted in Figure 5.

### 3.2. Analysis of Experiments and Significant Factors

As you know, a numerical factor is a variable that takes a certain value at some point in time and affects the optimization parameter. With regard to the problem being solved, the following factors have been tested that affect the objective function—the specific power of the electromagnetic core: rated apparent power $S_n$; rated voltage $U_n$; rated current frequency $f$; rated speed $n_n$; induction in the air gap $B_\delta$; saturation induction in the stator tooth $B_{nas}$; excitation winding voltage $U_v$. The rest of the design parameters were assigned fixed values, determined from the experience of designing electrical machines and the parameters of the selected electrical materials.

The initial selection of significant factors affecting the optimization parameter was carried out on the basis of a priori information about the factors and on the basis of the analysis of one-factor numerical experiments. These experiments were carried out at the first stage of complex parametric synthesis. To conduct and evaluate univariate experiments, the boundaries of the areas for determining factors were estimated. For inductor machines, a lot of a priori information has been accumulated about the boundaries of certain parameters. For example, single-phase inductor generators are manufactured with a capacity of 12–1500 kW, a frequency of 500–8000 Hz, a rotation frequency of 1500–3000 rpm and a voltage of up to 1500 V.

Further accurate selection of significant factors within the selected areas of their variation was carried out by the methods of mathematical statistics. A factor is considered significant if, when it changes within the scope of the factor definition, the change in the optimization parameter goes beyond the limits of the accuracy of determining the optimization parameter. The accuracy of determining the optimization parameter depends on the adequacy of the methodology for calculating the physical model, which is 88%–92%. Therefore, only those factors are taken into account where a change leads to a change in the optimization parameter by more than 12%.

Based on these criteria, the following four factors were selected as significant: rated current frequency $f$; nominal speed $n_n$; induction in the air gap $B_\delta$; saturation induction in the stator tooth $B_{nas}$.

Each combination of the levels of the selected factors is an external point in the factor space. It can be considered as a starting point for constructing an experiment and called the main (zero) level. The construction of an experiment plan is reduced to the selection of experimental points that are symmetrical about the zero level. To estimate the best area of the main level of factors, you can use the results of one-factor experiments (Figures 6–9).

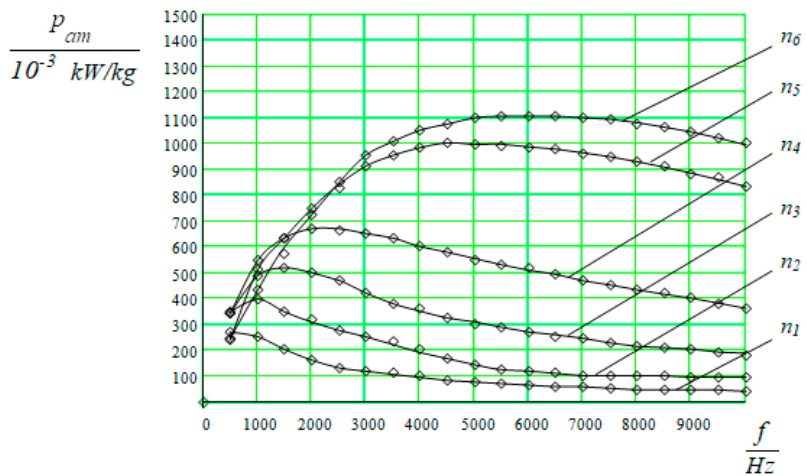

**Figure 6.** Dependences of the specific power $p_{am}$ on the current frequency at various generator rotation values: $n_1 = 10$ r/s; $n_2 = 25$ r/s; $n_3 = 50$ r/s; $n_4 = 100$ r/s; $n_5 = 300$ r/s; $n_6 = 400$ r/s.

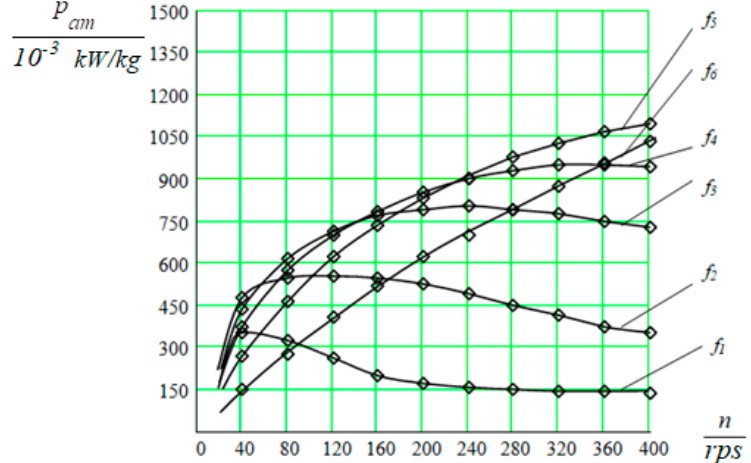

**Figure 7.** Dependences of the specific power $p_{am}$ on rotation frequency at different frequencies of current: $f_1 = 400$ Hz; $f_2 = 1000$ Hz; $f_3 = 2000$ Hz; $f_4 = 3000$ Hz; $f_5 = 5000$ Hz; $f_6 = 10,000$ Hz.

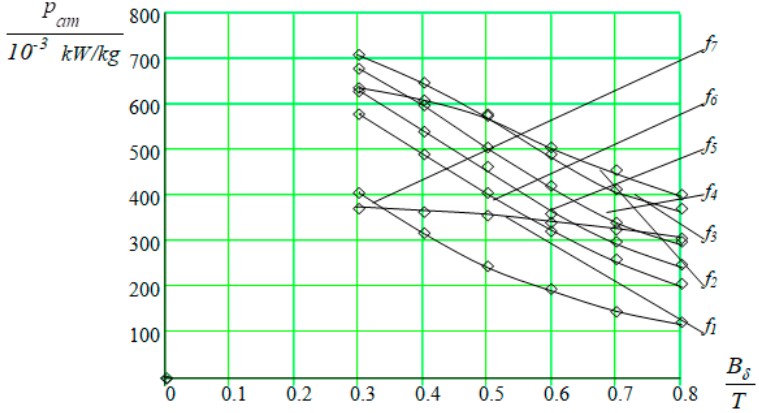

**Figure 8.** Dependence of the specific power $p_{am}$ on the value of induction in the air gap at a constant generator frequency: $f_1 = 400$ Hz; $f_2 = 1000$ Hz; $f_3 = 2000$ Hz; $f_4 = 3000$ Hz; $f_5 = 4000$ Hz; $f_6 = 5000$ Hz; $f_7 = 10,000$ Hz.

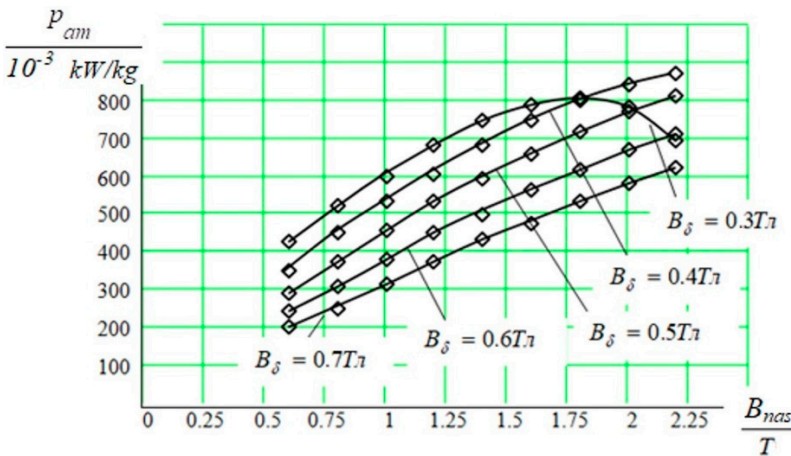

**Figure 9.** Dependences of the specific power $p_{am}$ on the induction of the saturation of the tooth and induction in the air gap.

They are shown as a family of curves at various fixed values of the other factor. The analysis of the optimization parameter maximum leads to the following results. The curves in Figure 6 show the dependence of the optimization parameter on the generator speed at a frequency of 5000 Hz, and you can set an approximately basic level of speed equal to 350 r/s.

From the graphs in Figure 6, it can be seen that with an increase in the speed of the generator drive motor, the optimization parameter increases in absolute units only up to a frequency of 5000 Hz and then decreases with an increase in frequency. Similar results are confirmed in Figure 7. Therefore, it is possible to take the frequency equal to 5000 Hz as the main level.

The curves in Figure 8 are at a frequency of 5000 Hz, and you can set the basic level of induction in the air gap equal to 0.4 Tesla.

Analyzing the curves in Figure 9, they can be taken as the main level of the limiting value of the saturation induction in the lower stator tooth, equal to 1.9 Tesla.

For every significant factor, it is necessary to set two symmetrical levels relative to the zero level of the factor. Let us take a higher value of the factor for the upper level and a lower one for the lower level.

The interval of variation of factors is called a certain number (its own for each factor), the addition of which to the main level gives the upper, and the subtraction gives the lower levels of the factor. To simplify the recording of the conditions of the experiment and the processing of experimental data, the coded value of the factors is used.

For factors with a continuous domain of definition, this is conducted using the transformation.

$$X_i = \frac{\widetilde{X}_i - \widetilde{X}_{io}}{I_i}. \tag{17}$$

where $X_i$ is the coded value of the factor; $\widetilde{X}_i$—the natural value of the factor; $\widetilde{X}_{io}$—the natural value of the main level of the factor; $I_i$—the factor variation interval; $i$—the factor number.

Then, according to Equation (15), the upper level of the factor in the coded designation will correspond to +1, the lower to 1 and the main one to zero. We denote the factor $B_{nas}$ by $X_1$, the factor $f$ by $X_2$, the factor $n_n$ by $X_3$, the factor $B_\delta$ by $X_4$.

The initial data for carrying out a full factorial experiment in the numerical and coded representation presented in Table 1.

**Table 1.** The initial data of a full factorial experiment.

| Factors | Levels | | | Variation Interval $I$ | Units |
|---|---|---|---|---|---|
| | $-1$ | $0$ | $+1$ | | |
| $X_1$ | 1.6 | 1.9 | 2.2 | 0.3 | T |
| $X_2$ | 3000 | 5000 | 7000 | 2000 | Hz |
| $X_3$ | 310 | 350 | 390 | 40 | rev/s |
| $X_4$ | 0.3 | 0.4 | 0.5 | 0.1 | T |

*3.3. Complete Factorial Experiment and Calculation Model Optimization*

In the general case, an experiment in which all possible combinations of factor levels are realized is called a complete factorial experiment. The experimental conditions are recorded in the form of a table. The lines correspond to different experiments and the columns to the values of the factors. Such tables are called experiment planning matrices. Each column in the scheduling matrix is called a column vector, and each row is called a row vector.

The choice of the number of parallel experiments is made to assess the variance of reproducibility. The analysis showed that for this study, the minimum number of parallel experiments, equal to two, is sufficient. The number of experiments required to implement all possible combinations of factor levels is found by the formula $N = 2^k$. Here $N$ is the number of numerical experiments; $k$ is the number of factors (in our case, 4); 2—the number of levels.

The matrix for planning a full factorial experiment for four factors in relation to the example under consideration is shown in Table 2. To move to the optimum point according to the results of planning the experiment, it is necessary to choose a linear model. The model is understood as the form of the response function y = $f(X_1, X_2, \ldots, X_k)$. Selecting a model means choosing the type of this function, writing down its equation.

**Table 2.** The matrix of the full factorial experiment.

| № | $X_0$ Code | $X_1$ Code | $X_2$ Code | $X_3$ Code | $X_4$ Code | $X_{1\times2}$ Code | $X_{1\times3}$ Code | $X_{1\times4}$ Code | $X_{2\times3}$ Code | $X_{2\times4}$ Code | $X_{3\times4}$ Code | $p_{am}$, kW/kg | |
|---|---|---|---|---|---|---|---|---|---|---|---|---|---|
| | | | | | | | | | | | | $Y_{v,1}$ | $Y_{v,2}$ |
| 1 | + | - | - | - | - | + | + | + | + | + | + | 1.267 | 1.365 |
| 2 | + | + | - | - | - | - | - | - | + | + | + | 1.113 | 1.164 |
| 3 | + | - | + | - | - | - | + | + | - | - | + | 1.628 | 1.607 |
| 4 | + | + | + | - | - | + | - | - | - | - | + | 1.410 | 1.464 |
| 5 | + | - | - | + | - | + | - | + | - | + | - | 1.327 | 1.270 |
| 6 | + | + | - | + | - | - | + | - | - | + | - | 1.079 | 1.068 |
| 7 | + | - | + | + | - | - | - | + | + | - | - | 1.812 | 1.789 |
| 8 | + | + | + | + | - | + | + | - | + | - | - | 1.547 | 1.530 |
| 9 | + | - | - | - | + | + | + | - | + | - | - | 0.936 | 0.948 |
| 10 | + | + | - | - | + | - | - | + | + | - | - | 1.461 | 1.178 |
| 11 | + | - | + | - | + | - | + | - | - | + | - | 1.346 | 1.421 |
| 12 | + | + | + | - | + | + | - | + | - | + | - | 1.116 | 1.115 |
| 13 | + | - | - | + | + | + | - | - | - | - | + | 0.906 | 0.953 |
| 14 | + | + | - | + | + | - | + | + | - | - | + | 1.180 | 1.141 |
| 15 | + | - | + | + | + | - | - | - | + | + | + | 0.923 | 0.919 |
| 16 | + | + | + | + | + | + | + | + | + | + | + | 1.144 | 1.186 |

It can be used to predict the response function in states that have not been studied experimentally. The main requirement for the model is the ability to predict the direction of further experiments with the required accuracy. A model that satisfies this requirement is called adequate. Most often, the model is sought in the form of power series—algebraic polynomials.

If we write out the polynomials for two factors, then they will differ in the maximum degrees of the variables included in them.

$$
\begin{aligned}
y &= b_0 \\
y &= b_0 + b_1 x_1 + b_2 x_2 \\
y &= b_0 + b_1 x_1 + b_2 x_2 + b_{12} x_1 x_2 + b_{11} x_1^2 + b_{22} x_2^2 \\
y &= b_0 + b_1 x_1 + b_2 x_2 + b_{12} x_1 x_2 + b_{11} x_1^2 + b_{22} x_2^2 + \\
&\quad b_{112} x_1^2 x_2 + b_{122} x_1 x_2^2 + b_{111} x_1^3 + b_{222} x_2^3
\end{aligned}
\tag{18}
$$

That is, there is a problem of approximating the unknown response function by a polynomial. The actual experiment is needed only to find the numerical values of the polynomial. The lower the degree of the polynomial for a given number of factors, the fewer the coefficients in it. Hence, it is necessary to find a polynomial that contains as few coefficients as possible but satisfies the requirements for the model.

The direction in which the model is good at predicting the fastest improvement in the optimization parameter is called the direction of the gradient. At the first stage of the search for the optimum, a polynomial of the first degree can be used. The condition of the analyticity of the response function guarantees the possibility that the linear model will be adequate in some arbitrary region.

At the next stage, a linear model is searched for in another subdomain. The cycle repeats until the movement along the gradient is no longer effective. This means getting into an area close to the optimum. The linear model is no longer needed here. Either the problem is solved by falling into an almost stationary region, or it is necessary to pass to polynomials of higher degrees in order to describe the optimum region in more detail.

The complete factorial method and the regression method known to the readers contain a new calculation model, which entailed a modification of the known methods for the optimal design of electric machines with magnetic commutation.

## 4. Discussion

After carrying out numerical experiments, statistical processing of the results is carried out, and the adequacy of the model is checked. At the same time, to assess the deviation of the optimization parameter $p_{am}$, kW/kg from the average value (Table 2), the reproducibility variance is calculated according to the data of parallel observations of the planning matrix plan at each point. After that, the homogeneity of dispersions is checked according to the Cochran test. The significance of the regression coefficients in the mathematical model is determined by the Student's criterion. Further, according to Fisher's criterion, the adequacy of the model is checked. If the model is adequate, then you can proceed to a steep ascent.

A steep ascent is carried out according to the Box–Wilson method. It implements movement in the direction of the gradient of the response function. The gradient is given by the partial derivatives, and the partial derivatives of the response function are estimated by the regression coefficients. The independent variables (factors) are changed in proportion to the values of the regression coefficients, taking into account their signs. The components of the gradient are obtained by multiplying the regression coefficients by the variation intervals for each factor. A series of subsequent steps in the direction of the gradient is calculated by successively adding to the base level of the factors' values proportional to the components of the gradient.

Movement along the gradient is considered effective if the implementation of numerical experiments at the stage of steep ascent leads to an improvement in the value of the optimization parameter compared to the best result in the matrix. If the optimum region is

reached, then a decision is made to end the experiments; otherwise, a linear plan for the next cycle is set, and the study is repeated.

The obtained design results in a single-phase inductor generator of axial-radial configuration (Figure 6) in comparison with an inductor generator of classical design with a drum rotor are shown in Table 3. The generators comply with the design specifications: apparent power $S$ = 111 kVA; rated voltage $U_n$ = 750 V; rated current $I_n$ = 148 A; power factor $\cos\varphi_n$ = 0.9.

**Table 3.** Comparative characteristics of inductor generators.

| Inductor Generator | Current Frequency Hz | Rotation Frequency rpm | Diameter m | Length m | Specific Power kW/kg | Efficiency |
|---|---|---|---|---|---|---|
| Classic Drum Rotor Design | 2700 | 3000 | 0.740 | 0.432 | 0.10 | 0.89 |
| Axial-Radial Configuration (Project Without Parametric Synthesis) | 2700 | 3000 | 0.836 | 0.366 | 0.43 | 0.88 |
| Axial-Radial Configuration (Parametric Synthesis) | 5700 | 22,200 | 0.338 | 0.355 | 1.89 | 0.93 |

Designing an inductor generator of axial-radial configuration with the frequency of a classical inductor generator, without the use of parametric synthesis, gives a gain in specific power only 4.4 times. Analysis of the results of parametric synthesis of a modular inductor machine according to the proposed method shows the possibility of obtaining an optimal generator with a specific power exceeding the specific power of an inductor generator of a classical design with a drum rotor by more than 18 times. The use of parametric synthesis in design gives a fourfold increase in the energy density in a modular machine. The results obtained indicate the undeniable advantages of modular machines in comparison with machines with a drum rotor.

The authors modify known methods for the optimal design of a new research object. This new research object is magnetic commutated modular machines. A new computational model was applied, which made it possible to establish links between the design task, the geometry of the machine and the characteristics of the new object of study (Equations (6)–(11), Figures 3–9). This is the usefulness of the method. As a result of the modification, a complex parametric synthesis of a new research object has been developed.

The method is needed to create a new object of research with a minimum mass and maximum power. The known methods used for the design of known cylindrical electrical machines cannot be applied in this case. Cylindrical machines have a distributed electrical and lumped magnetic system. Additionally, modular machines with magnetic commutation have a distributed magnetic and lumped electrical system. A new approach is required to establish links between the design assignment and the geometry of the machine.

The originality and novelty of the method lie in the fact that for the new object of research, the authors also developed a new computational model based on the main computational Equation (3). The new computational model allows you to establish links between the design task, the geometry of the machine and its characteristics (Equations (6)–(11), Figures 3–9). This entailed a modification of optimal design methods applied to a new object of research.

## 5. Conclusions

The design of electrical machines consists of an interconnected sequence of stages and operations, during which there is a gradual detailing of the system being developed and a more complicated mathematical description of the processes is applied. The specific weight of the use of research methods, as the design deepens, will be mixed with parametric synthesis and analysis in order to obtain the minimum cost, at maximum power, within the specified masses and dimensions of the electric machine.

A block diagram of a complex parametric synthesis for the design of energy-efficient magnetically commutated modular electrical machines is proposed. The methods of analysis and synthesis used to solve these problems are considered, as well as mathematical models, computational algorithms and programs.

On the example of the study of an inductor electric generator of axial-radial configuration, new approaches to the formulation and solution of typical problems of analysis and synthesis of modular electric machines are shown. The use of complex parametric synthesis makes it possible to significantly reduce the masses of the designed modular machines in comparison with drum-type inductor machines of the same power.

**6. Patents**

Patents related to the results of the work indicated in this manuscript:

1. Afonin A., German-Galkin S., Cierzniecki P., Hrynkiewicz J., Kramarz W., Szymczak P. Modular reluctance machine. PCT Int. Public Nomber WOo 1/03270.Al. Int. Public. Date 11 January 2001. Priority data 22 May 1999.
2. Шайтор М.М., АфонінА.О., Рясков Ю.І. Електромеханічна система аксіально-радіальної конфігурації. ПатентUA 49630. Опубликовано: 15 December 2003. База патентів України. Available online: URL uapatents.com (accessed on 25 May 2021).
3. Шайтор М.М., Рясков Ю.І., Березовенко О.В. Магнітокомутуючий двомережевий генератор зсамозбудженням. ПатентUA 62802. Опубликовано: 15 December 2003. База патентів України. Available online: URL uapatents.com (accessed on 25 May 2021).
4. Шайтор М.М., Рясков Ю.І., ШевцовЕ.І. Трифазний двигунаксіально-радіальної конфігурації. ПатентUA 69069. Опубликовано: 16 August 2004. База патентів України. Available online: URL uapatents.com (accessed on 25 May 2021).
5. Шайтор М.М., Рясков Ю.І., ЖидковВ.О. Безшатуновий безщітковий генераторний агрегат. ПатентUA 68262. Опубликовано: 15.07.2004. База патентів України. Available online: URL uapatents.com (accessed on 25 May 2021).
6. Шайтор М.М., Рясков Ю.І. Магнітокомутуюча дискова машина з кігтеподібниминд уктором. ПатентUA 62556. Опубликовано: 15 December 2003. База патентів України. Available online: URL uapatents.com (accessed on 25 May 2021).
7. German-Galkin S., Hrynkiewicz J. Uklad modulu elektromechanicznego maszyny elektrycznej. Pat. UP RP z 20 December 2013r. Nr P.389456. Zglosz, pat. 03 November 2009.
8. БормотавА.В., Герман-ГалкинС.Г., Загашвили Ю.В., ЛебедевВ.В. Модульная электрическая машина. ПатентRU 2,510,121 C2, 20 March 2014.

**Author Contributions:** Conceptualization, B.Y. and N.S.; methodology, N.S.; software, N.S.; validation, B.Y., N.S. and M.K.; formal analysis, B.Y.; investigation, N.S.; resources, M.K.; data curation, B.Y.; writing—original draft preparation, M.K.; writing—review and editing, B.Y.; visualization, N.S.; supervision, N.S.; project administration, N.S.; funding acquisition, M.K. All authors have read and agreed to the published version of the manuscript.

**Funding:** This research received no external funding.

**Institutional Review Board Statement:** Not applicable.

**Informed Consent Statement:** Not applicable.

**Data Availability Statement:** Not applicable.

**Acknowledgments:** The authors would like to thank the Slovak Grant Agency VEGA 1/0389/18 and project VEGA 1/0201/21.

**Conflicts of Interest:** The authors declare no conflict of interest.

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
