# Peer review of "Analysis and Synthesis in the Design of Magnetic Switching Electric Machines"

_actuators, doi:10.3390/act10070164_

Round 1

Reviewer 1 Report

The presented article deals with design of specified electric machines. The main focus is in the mathematical analysis of design and parameters to reduce physical dimensions and improve power of such devices. This topic is nowadays very actual.

Formal comments:

If new sentence begins after formula, please consider to change "," with "."

The quality of Figure 2, Figure 4 must be improved - size and details visibility.

Figure 8: Please consider the improving the style of curves in graph (colors, etc.) for better reading. (The saqme style may be used for other graphs).

Possible improvements:

The results, own design is fully supported by mathematical apparate. In Materials and Methods section, it would be interesting to extend and support (by pseudocodes or mathematically) the blocks in Figure 1.

Page 3, bottom (line 111): ...options that are not suitable for life... - please explain this sentence.

Reviewer 2 Report

You show your opinions on the research methodology for modular actuator. 

However, your analysis and synthesis method for optimizing electromagnetic machine do not have originality and novelty. There are so many optimizing methods to have better performance. But I am not sure your method is more useful than others. 

First, you should focus your introduction part on your background of research. Why is your method needed? and What is your originality?

The full factorial method and regression method you used for optimization is not new to readers any more. 

Reviewer 3 Report

There are too much typos and inscriptions. And the sentences are not specific and the explanation is ambiguous.

ex) “By their design, they are referred to modular machines, and by the principle of operation - to synchronous reluctance-inductor machines”(row 48 of page 2); What it means?

I cannot understand what the magnetically commutated inductor machine in this paper. Does it mean induction generator? – Fig 2 doesn’t seem as Induction machine. Ref [14-16] which shows what it means is not accessible, so you need to includes additional cross-section view or 3D model to explain the configuration of the machine.
